# The Influence of Filler Size and Crosslinking Degree of Polymers on Mullins Effect in Filled NR/BR Composites

**DOI:** 10.3390/polym13142284

**Published:** 2021-07-12

**Authors:** Miaomiao Qian, Bo Zou, Zhixiao Chen, Weimin Huang, Xiaofeng Wang, Bin Tang, Qingtao Liu, Yanchao Zhu

**Affiliations:** 1College of Chemistry, Jilin University, Changchun 130012, China; qianmm19@mails.jlu.edu.cn (M.Q.); dr.z@jlu.edu.cn (B.Z.); huangwm@jlu.edu.cn (W.H.); 2State Key Laboratory of Inorganic Synthesis & Preparative Chemistry, College of Chemistry, Jilin University, Changchun 130012, China; zhixiaocheng18@mails.jlu.edu.cn (Z.C.); wangxf103@jlu.edu.cn (X.W.); 3Institute for Frontier Materials, Deakin University, Melbourne/Geelong, VIC 3216, Australia; bin.tang@deakin.edu.au; 4Monash Institute of Pharmaceutical Sciences, Monash University, Melbourne, VIC 3800, Australia

**Keywords:** Mullins effect, energy dissipation, composite structure

## Abstract

Two factors, the crosslinking degree of the matrix (*ν*) and the size of the filler (*Sz*), have significant impact on the Mullins effect of filled elastomers. Herein, the result. of the two factors on Mullins effect is systematically investigated by adjusting the crosslinking degree of the matrix via adding maleic anhydride into a rubber matrix and controlling the particle size of the filler via ball milling. The dissipation ratios (the ratio of energy dissipation to input strain energy) of different filled natural rubber/butadiene rubber (NR/BR) elastomer composites are evaluated as a function of the maximum strain in cyclic loading (*ε_m_*). The dissipation ratios show a linear relationship with the increase of *ε_m_* within the test range, and they depend on the composite composition (*ν* and *Sz*). With the increase of *ν*, the dissipation ratios decrease with similar slope, and this is compared with the dissipation ratios increase which more steeply with the increase in *Sz*. This is further confirmed through a simulation that composites with larger particle size show a higher strain energy density when the strain level increases from 25% to 35%. The characteristic dependence of the dissipation ratios on *ν* and *Sz* is expected to reflect the Mullins effect with mathematical expression to improve engineering performance or prevent failure of rubber products.

## 1. Introduction

Elastomeric materials with fillers have a broad range of industrial applications because of their unique properties, such as high tensile strength, deformability, and toughness. However, the addition of fillers also causes significant inelastic behavior [1,2,3]. When composites are stretched from their virgin state, unloaded and then reloaded, the stress required on reloading is lower than that during the initial loading; this stress softening phenomenon called the Mullins effect [4,5]. The Mullins effect is generally attributed to a continuous damage process; however, the damage mechanisms under a large cyclic applied stress are still not fully understood [6]. The Mullins effect is involved in both the engineering performance and failure of rubber products, therefore, a comprehensive understanding of the Mullins effect is important [7,8,9].

The possible physical mechanisms and mathematical models of the Mullins effect have attracted great attentions from researchers. Mullins et al. [10,11] proposed that filled elastomers have amorphous micro-structures consisting of hard and soft phases. The hard phase transformed into the soft phase during the deformation process, which led to the Mullins effect. This concept has been used to explain the Mullins effect in many filled elastomers [12,13,14,15]. The Mullins effect is also proposed to be based on continuum damage mechanics, and strain constitutive equations coupled with damage was used to describe this effect [16,17,18,19,20]. Other physical mechanisms proposed to explain the Mullins effect include bond rupture [21,22,23,24], molecular slip at the matrix–filler interface [25,26], breakdown of aggregates and agglomerates of filler particles [27,28], and disentanglement [29,30]. It is believed that the mechanisms vary with the nature of the filler and polymer of the filled elastomers [31]. Li et al. divided the energy loss accompanying the Mullins effect into recovery hysteresis (*E_rh_*) and softening parts (*E_s_*). They found that both *E_rh_* and *E_s_* were dependent on nanocomposite structure (filler volume fraction and crosslinking degree) [32]. Strain-induced light emission from mechanoluminescent cross-linkers in silica-filled poly(dimethylsiloxane) has been recently used to reveal the covalent bond scission which contributes to the Mullins effect [33]. However, the experimental works mentioned above are complex to achieve in composites, which hinder their wide application. Therefore, establishing a simple method to evaluate the influence of cross-linking degree of matrix and filler size on Mullins effect would promote better understanding of the Mullins effect, and may eventually improve the performance of the products.

Based on Green’s equation ($W=\int_{0}^{\varepsilon_{ij}}\sigma_{ij}d\varepsilon_{ij}$), it is feasible to quantitatively analyze the Mullins effect based on the energy change rather than the stress difference. It is worth noting that the strain interval should be the same when calculating the energy of different loading paths. There is an energy reduction when a filled elastomeric material is subjected to cyclic loading with tension from its initial (virgin) natural configuration, whereas the largest reduction is observed in the first and second cycles. Therefore, it is reasonable to evaluate the Mullins softening with the energy difference corresponding to the same strain between the first and the second cycle. It has been reported that the exact mechanisms of Mullins effect varied with the nature of the polymer and filler of the system. If the energy difference also varies systematically with the polymer and filler of the system, the energy difference would provide a criterion for evaluating the degree of Mullins effect in the experimental range.

In this study, the influence of two key factors, crosslinking degree of matrix and size of filler, on the Mullins effect were systematically investigated. The two key factors were purposely controlled, cross-linking degree was adjusted through via adding maleic anhydride into rubber matrix and particle size of the filler was changed via ball milling. The dissipation ratios during cyclic uniaxial tensile tests were evaluated as functions of the maximum strain in cyclic loading and they showed linear relationships with the increase of the *ε_m_* within the test range. Crosslinking degree determines the value of the dissipation ratios, but the size of the filler affects the trend of the dissipation ratios with *ε_m_*, which indicates crosslinking degree and size of filler have different contributions to the Mullins effect. These results provide new insights for designing filled elastomeric materials and predict their behavior.

## 2. Materials and Methods

### 2.1. Materials

Natural rubber (RSS1) consists of cis-1,4-polyisoprene with a Mooney viscosity of 79.9 at ML (1 + 4) at 100 °C, and butadiene rubber (BR9000) consists cis-1,4-polybutadiene with a Mooney viscosity of 50 at ML (1 + 4) at 100 °C. Both natural rubber and butadiene rubber were from Shanghai Dukang Co., Ltd., (Shanghai, China). The pyrolytic rice husk ash containing biochar and silica (Trade Name as SiCB) were produced by Jilin Kaiyu Biomass Development and Utilization Co., Ltd. (Changchun, China). Milled SiCB (MSiCB) were prepared according to reported procedure using the XQM-2 planetary ball mill (Tianchuang Co., Changsha, China) [34], and the milling parameters of SiCB are described in Appendix A. Maleic anhydride (MA), stearic acid, zinc oxide (ZnO), and sulfur (S) in analytical grades were purchased from Sinopharm Chemical Reagent Co., Ltd. (Shanghai, China). All the other chemicals including *N*-1,3-dimethylbutyl-*N*′-phenyl-P-phenylenediamine (antioxidant 4020), poly(1,2-dihydro-2,2,4-trimethyl-quinoline) (antioxidant RD), wax, *N*-tertbutylbenzothiazole-2-sulphenamide (accelerator NS), dicumyl peroxide (DCP, 98%), and aromatic hydrocarbon oil (DAE) in chemical grade were supplied by Beijing Chemical Works. (Beijing, China) and used without further purification.

### 2.2. Preparation of Filled Elastomer Composites

Natural rubber/butadiene rubber (NR/BR) were chosen as elastomer matrix due to activity in chemistry of NR which provides the chance for graft of maleic anhydride and low thermogenesis of BR which reduces energy loss. Mechanical properties of NR/BR materials are shown in Appendix A. The elastomer composites were prepared using a laboratory-sized internal mixer (KY-3220C-1L, Dongguan Kaiyan Machinery Equipment Factory, Dongguan, China) according to the formulations listed in previous work [35]. NR and MA were first mixed in an internal mixer at 40 rpm and 100 °C for 10 min to incorporate MA onto the main chain of NR. Then pre-weighted filler, BR and rubber additives were added and blended for another 13 min. Finally, the obtained composites were hot-pressed at 150 °C and 20 MPa. The prepared samples were named as SiCB-MNR_x_/BR, with the numbers x representing the corresponding loading of MA with a unit of phr. SiCB was milled under different conditions (M_1_SiCB and M_2_SiCB) and were used as the fillers. The as-prepared milled SiCB mixed with NR/BR to prepare M_1_SiCB-NR/BR and M_2_SiCB-NR/BR. The detailed sample information is listed in Table 1.

### 2.3. Characterization

Fourier transform infrared (FTIR) spectra were collected on a Bruker V70 FTIR spectrometer (Bruker, Karlsruhe, Germany) using attenuated total reflection (ATR) mode in order to substantiate the interaction between NR and MA. Particle size distribution of the fillers were measured by the laser scattering particle size analyzer (BT-9300ST, Baite Instrument, Dandong, China) using water as the dispersant. The particle morphology and cryogenic fracture surfaces of composites were observed using scanning electron microscope (SEM, SU8020, Hitachi & SEM, JSM-6700F, Jeol, Tokyo, Japan). The crosslinking density [*ν*]*_es_* was determined from equilibrium swelling experiments with toluene. Rubber samples (around 1 g) were weighed in 50 mL vials (*m*_0_) and immersed in 20 mL of solvent to reach equilibrium swelling (72 h). The solvent was removed from the sample surface with filter paper and the sample was (*m*_1_) and finally dried at 80 °C in an oven for 48 h until a constant weight was reached (*m*_2_). The crosslinking density was calculated according to the Flory–Rehner expression: [36,37]
(1)$${\left[\nu \right]}_{es}=\frac{\ln\left(1-V_{R}\right)+V_{R}+\chi V_{R}^{2}}{2V_{s}\left(0.5V_{R}-V_{R}^{\frac{1}{3}}\right)}\;\mathrm{with}\;V_{R}=\frac{m_{2}}{m_{2}+\left(m_{1}-m_{2}\right)\frac{\rho_{R}}{\rho_{S}}}\;$$
(2)$$\;\rho_{R}=0.45\rho_{NR}+0.55\rho_{BR}\;$$
where *V_R_* is the volume fraction of rubber in swollen sample, vs. is the molar volume of toluene (106.3 mL/mol), *χ* is the NR/BR–toluene interaction parameter (here, *χ* = 0.534), $\rho_{R}$ and $\rho_{S}$ are the density of rubber ($\rho_{NR}$: 0.930 g/mL; $\rho_{BR}$: 1.930 g/mL) and toluene (toluene: 0.866 g/mL), respectively.

The crosslinking density [*ν*]*_ts_* was determined from the stress–strain curves via the Mooney–Rivlin approach [38,39,40]. The crosslinking density was subsequently calculated from *C*_1_.
(3)$$\sigma =2\left(C_{1}+\frac{C_{2}}{\lambda }\right)\left(\lambda -\frac{1}{\lambda^{2}}\;\right)\;with\;\lambda =1+Χ\varepsilon \;$$
(4)$${\left[\nu \right]}_{ts}=\frac{2C_{1}}{k_{B}T}\;$$
where *σ* is the true stress, which is measured in the strained state, *C*_1_ and *C*_2_ are characteristic parameters of the vulcanized rubber, representing effects of chemical crosslinking and entanglements, respectively. *λ* is the extension ratio, *X* is the strain amplification factor (*X* = 1 for gum rubber), *ε* is the engineering strain, *k_B_* is the Boltzmann constant (1.38 × 10^−23^ m^2^ kg s^−2^ K^−1^), and *T* is the absolute temperature.

Uniaxial cyclic tension test was conducted on the CMT-20 testing machine equipped with an extensometer (Liangong Testing Technology Co., Ltd., Jinan, China). The rectangular specimens with a dimension of 70 mm × 12.5 mm × 3.5 mm were stretched with a constant strain rate of 0.025 s^−1^ (*l*_0_ = 40.0 mm).

To investigate the Mullins softening, the specimens were stretched using various the maximum strain in cyclic loading (*ε_m_*, *ε_m_* varied from 0.5 to 3.0 for the samples). The measurements were repeated at least three times for each specimen to confirm satisfactory reproducibility.

## 3. Results and Discussion

### 3.1. Characterization of the Fillers

The particle size of filler is one of the important factors that substantially affect the performances of filled rubber composites [41,42]. Figure 1a shows particle size distribution of fillers of the as-prepared composites. The size distribution of SiCB particles ranged from tens of nanometers to 70 μm broadly. After ball milling treatment, the obtained M_1_SiCB showed a reduced size and a narrower size distribution ranging from a few nanometers to 20 μm. When the milling speed was further increased, the obtained particles shifted to a smaller size range, and at the same time the particles started to form agglomerates ranged from 10 to 60 μm due to the excessive surface energy (Figure 1a).

The morphologies of the obtained fillers were observed by SEM. SiCB showed large irregular shape with bright particles distributed over the surface of the primary particles, suggesting the presence of two different materials [43] (Figure 1b). After ball milling, the as-prepared M_1_SiCB showed reduced particle size than SiCB (Figure 1c), while big agglomerates formed in M_2_SiCBs with uneven surfaces (Figure 1d), which are consistent with the particle size analyzer result. These three fillers with same composition but different size range were further used to study filler size effect on viscoelasticity in the following session.

### 3.2. Crosslinking Density of SiCB-MNR_x_/BR Composites

Crosslinking density (number of crosslinking bonds in a crosslinked polymer) is one of the main variables affecting the properties of vulcanized rubbers. Crosslinking densities of the SiCB-MNR_x_/BR composites with various MA contents were determined by equilibrium swelling (Table 2). With an increasing MA contents, the crosslinking density determined from equilibrium swelling ([*ν*]*_es_*) increased gradually, indicating changes of the rubber matrix after adding MA (confirmed by FTIR spectra in Appendix A). The crosslinking density determined from the stress–strain curve ([*ν*]*_ts_*) was also obtained, which was larger than [*ν*]*_es_*. This may be because [*ν*]*_ts_* includes both the chemical crosslinking and the temporarily trapped chain entanglements [44,45,46].

### 3.3. Properties of the Filled NR/BR Composites

The effect of the filler particle size and crosslinking degree of polymer matrix on the filler reinforce performance was evaluated in detail. Figure 2a displays stress–strain curves of NR/BR composites with different crosslinking. Among all the composites, SiCB-NR/BR showed the lowest stress of 4.9 MPa upon breakage, which could be attributed to the poor compatibility between filler SiCB and rubber matrix. However, the stress increased to 6.2 MPa after the inclusion of MA (1 phr). The crosslinking degree of polymer matrix improved with an increase in the MA content up to 3 phr, leading to increased tensile strength as shown in Figure 2a.

Stress–strain curves of SiCB-NR/BR composites with different filler particle sizes are shown in Figure 2b. Compared to SiCB, the milled SiCB has a smaller size with an enhanced reinforcing performance. This is because the filler with a small size provided a large contact area with the rubber matrix and offered more sites to share the stress loaded on the molecular chains. The reinforcing performance of M_2_SiCB was lower than M_1_SiCB, which may be attributed to the non-uniform size distribution of M_2_SiCB. Compared with M_1_SiCB-NR/BR and M_2_SiCB-NR/BR, the stress of SiCB-NR/BR increased quickly at the initial stage, which arose from the large particle size of filler acting as hindrance during tensile testing. The stress–strain curve gradually becomes flattened as further stretching the SiCB-NR/BR sample due to the rupture of filler clusters [47].

The uniformity of the filler dispersion in the matrix also played an important role in the composite properties. The fracture surfaces of the composites were observed by SEM as shown in Figure 3. The aggregation of filler particles and cavities in the matrix were observed on the fracture surface of SiCB-NR/BR (Figure 3a), which indicated the weak filler–rubber interaction. In contrast, no obvious phase separation between the filler and polymer matrix was observed for the M_1_SiCB-NR/BR (Figure 3b), which indicates that small particle size and uniform dispersion of M_1_-SiCB in the rubber matrix could improve interfacial compatibility. This, in turn, should result in the improved reinforcement in the mechanical property of composites, consistent with the previous report [34]. Small biochar block and rough surface were observed on the fracture surface of SiCB-MNR_3_/BR (Figure 3c), which indicated enhanced interfacial adhesion between SiCB and MNR/BR polymer matrix due to the presence of MA.

### 3.4. Mullins Effect of the Filled NR/BR Composites

To study the Mullins effect of the filled NR/BR composites, uniaxial cyclic tensile tests were conducted at various strains from 0.5 to 3.0. As shown in Figure 4, a significant stress softening was observed for SiCB-NR/BR, SiCB-MNR_3_/BR and M_1_SiCB-NR/BR in the first two tensile loop. The stress required on reloading was lower than that during the initial loading in the regime of *ε* < *ε_m_* for all the composites (Appendix A). This result indicates that all composites exhibited Mullins effect which is considered a damage mechanism of rubber materials.

In order to further investigate the Mullins effect, three basic parameters are defined, namely, stored elastic energy in loading and reloading processes (*W*_0_ and *W_r_*), and the released energy in the unloading process (*W_u_*) as shown in Figure 5. Four derived parameters have also been defined based on *W*_0_, *Wr*, and *W_u_*. The list of notations has been showed in Table A1.

The first two derived parameters, Energy dissipation *D* is obtained from the area enclosed by the loading–unloading or loading–reloading curves as shown in Figure 5. *D_u_* and *D_r_* are defined by the following equations.
(5)$$D_{u}=W_{0}-W_{u}\;$$
(6)$$D_{r}=W_{0}-W_{r}\;$$
where *D_u_* represents the difference between stored elastic energy and released energy in each loading–unloading cycle. *D_r_* represents the difference of stored elastic energies in loading–reloading cycles.

Figure 6 shows the energy dissipation of SiCB-MNR_x_/BR composites with different crosslinking densities at different strain *ε* during the unloading (*D_u_*) and reloading (*D_r_*) processes. It can be seen that both *D_u_* and *D_r_* increased with the increase of *ε_m_* for all composites. *D_u_* was larger than *D_r_* at the same strain, which indicates that stress in reloading was larger than that in unloading at the same *ε*. The energy difference is indicative of the viscoelasticity in the filled elastomers [5]. Li et al. investigated the energy dissipation accompanying the Mullins effect in virgin loading–unloading process. They divided the energy dissipation into recovery hysteresis and softening stages, and suggested the energy loss associated with hysteretic recovery was mainly determined by the microscopic strain of the rubber phase, while the energy loss associated with softening was involved in both the rubber and filler phases [32]. Compared with the virgin loading–unloading process, energy dissipation in loading–reloading processes was defined as the permanent hysteresis energy, which was regarded as irreversible energy dissipation [33].

It should be considered that the input energy by loading (*W*_0_) increases with the increase of *ε_m_*, resulting in increased *D* at the same time [48]. In order to compare energy dissipation *D* with consideration of *W*_0_ among the deformations under various degrees of strain, a derived parameter ‘dissipation factor’ (*Δ*) can be defined as the ratio of energy dissipation to the stored elastic energy of loading (input elastic energy):(7)Δ i =DiW0 i=u,r 

The filled elastomers exhibit a significantly lower stress on reloading than that on the virgin loading under the previously applied maximum strain. Larger stress difference indicates more energy dissipation resulting from the damage between the first and the second loading paths. Therefore, *Δ_r_* associated with loading–reloading process could be as a measure of the degree of softening effect. Large *Δ_r_* indicates high energy dissipation, which could reflect high manifest degree of the Mullins effect. Assuming that the loss factor under the experimental conditions is a function of elastic matrix crosslink density (*ν*), particle size (*Sz*) and strain (*ε*), it can be described by the following equation:(8)$$\Delta_{r}\;=\xi \;\left(\nu ,Sz,\varepsilon \right)$$
(9)$$\Delta_{r}\;=\varphi \left(\nu \right)$$
(10)$$\Delta_{r}\;=\phi \left(Sz\right)$$

### 3.5. Influences of Crosslinking Degree of Polymer Matrix on the Mullins Effect

Figure 7a displays *Δ_r_* as a function of *ε_m_* for SiCB-MNR_x_/BR composites with different crosslinking densities. It can be seen that *Δ_r_* increased with increasing the strain for SiCB-MNR_x_/BR composites. It is worth mentioning that all the SiCB-MNR_x_/BR composites showed a very close slope of the fitting curve, which was around 7.6. This result may suggest that when SiCB-MNR_x_/BR composite specimens completed the first two stretching cycles under the same maximum strain, the ratio of the energy dissipation to the total stored energy of the composite showed a simple linear relationship with the change of the *ε_m_*. Since the SiCB-MNR_x_/BR composites were prepared with the same filler under the same conditions, the only difference between these composites was the crosslinking degree of the elastomer. This finding reveals that the softening depends on the crosslinking degrees of the rubber when the other conditions are the same. Furthermore, higher crosslinking density resulted in the weaker softening degree, this difference had no correlation with the change of *ε_m_* of the specimen. It was also found that higher *ν* would lead to smaller *Δ_r_*, which could be ascribed to increased chemically crosslinked network; higher molecular weight polymer network provided more extra reversible deformation, thus there would be less energy dissipation. Zhong et al. [49,50] proposed that increasing the elastomer crosslinking degree would result in chain entanglement, slippage, and twisting more likely to occur, therefore leading to the magnitude of strain energy change. However, this strain energy change would also be averaged over the different strains. Therefore, for the samples with the same *Sz*, *ν* only affected the value of *Δ_r_*, but did not affect the trend of *Δ_r_* as a function of *ε_m_*$\left(\frac{d\Delta_{r}}{d\varepsilon }=\frac{d\varphi \left(\nu \right)}{d\varepsilon }=0\;\right)$.

On the other hand, *Δ_u_* only reflects the energy dissipation in first loading–unloading process, this is in agreement with the reported theory that the main Mullins effect is represented by the difference between the first and the second loading–unloading runs. Therefore, to compare *Δ_u_* during the loading–unloading processes among the as-prepared composites, the difference of the dissipation factors (*Ω* (Δ1*_u_ -*Δ2*_u_*)) between the first and second loading–unloading runs was used to reflect the degree of the softening effect. A larger *Ω* indicates a higher energy dissipation, which would reflect a higher manifest degree of the Mullins effect. The *Ω* value as a function of *ε_m_* for SiCB-MNRx/BR composites was shown in Figure 7b. It can be seen that *Ω* increased with increasing the strain for SiCB-MNR_x_/BR composites, which was *ν*-independent. For SiCB-MNR_x_/BR composites, *Ω* behaves very similarly to *Δ_r_* under different strains, as shown in Figure 7c.

### 3.6. Influences of the Filler Particle Size on the Mullins Effect

The influences of the filler particle size on Mullins effect of the SiCB-NR/BR composites was also evaluated in detail. As shown in Figure 8, for the same SiCB-NR/BR sample, *Δ_r_* and *Ω* increased with increasing the strain. It was also observed that the slope of the fitting curve of the three samples follow the order of SiCB-NR/BR > M_2_SiCB-NR/BR > M_1_SiCB-NR/BR, which corresponds to the filler size order of SiCB > M_2_SiCB > M_1_SiCB. Since the samples were prepared under the same conditions, we suggest that the slop differences of the three samples are ascribed to the particle size of the filler. The reduced filler particle size resulted in a decreased slope, which indicates that the trend of *Δ_r_* as a function of *ε_m_* is *Sz*-dependent $\frac{d\Delta_{r}}{d\varepsilon }=\frac{d\phi \left(Sz\right)}{d\varepsilon }=b,$ (b is a constant). The average particle size of the filler SiCB in the composite matrix would result in the molecular network undergoing a high degree of structural breakdown during the cyclic deformation, which could lead to increased energy loss and more obvious softening behavior. These results reveal that *ν* and *Sz* have different influence on the Mullins effect, *ν* determines the intercept of *Δ_r_* and *Ω* (Figure 7c), while *Sz* affects the slope of *Δ_r_* and *Ω* (Figure 8c) with *ε_m_*. Therefore, both particle size and crosslinking degree are key factors that determines the Mullins effect in filled rubber composites.

Varying fillers particle size is limited due to the nature of filler SiCB, therefore, the particles size effect on strain energy density are further investigated by simulation. The Mooney–Rivlin model was used to simulate the particle size effect on the strain energy density of the composite samples via finite element analysis coded with ABAQUS, which was carried out at the different strain. Uniaxial cyclic test data was used as the input source to calculate the coefficients. Figure 9 shows the strain energy density distribution profiles of model NR/BR composites, where fillers are schematized by rigid body balls with different radii (r = 0.1, 0.2, 0.3 and 0.4 mm) at the strain level of 25% and 35%, respectively. Composites with larger particle size show a higher strain energy density of composites when the strain level increases from 25% to 35%.

## 4. Conclusions

In summary, *Δ_r_* and *Ω* associated with energy dissipation from different composites subjected to uniaxial cyclic tensile testing were used for the first time to evaluate the effect of filler particle size and polymers’ crosslinking degree on the Mullins effect. The study was performed by evaluating the relationship between *Δ_r_*, *Ω* of the different composites and the maximum strain. When the particle size of filler was the same, increased crosslinking degree led to a reduced Mullins effect of the composites. In contrast, filler with larger particle size promoted the Mullins effect when other conditions were the same. *Δ_r_* and *Ω* could be the effective factors to describe the Mullins effect, which makes it possible to understand the softening performance of rubber composites caused by the filler size or crosslinking degree of matrix in practice.

## Figures and Tables

**Figure 1 polymers-13-02284-f001:**
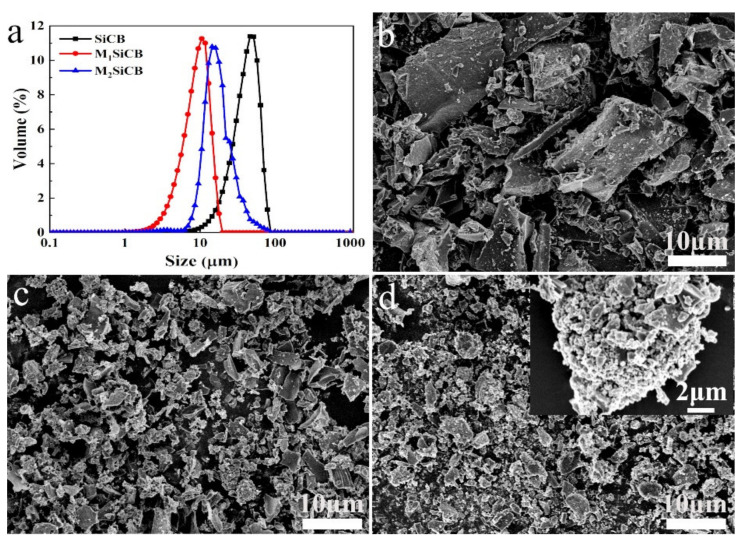
(**a**) The particle size distribution of fillers, SEM of (**b**) SiCB, (**c**) M_1_SiCB and (**d**) M_2_SiCB.

**Figure 2 polymers-13-02284-f002:**
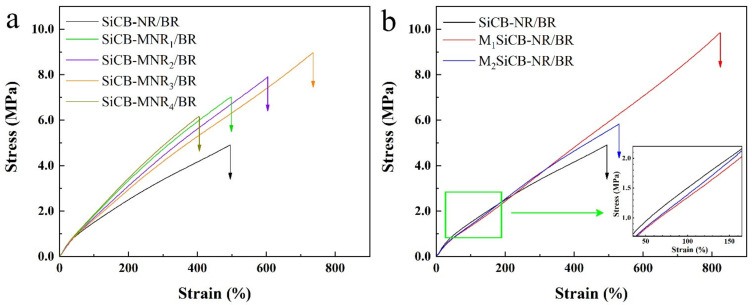
Representative stress–strain curves of the filled NR/BR composites: (**a**) with different crosslinking densities and (**b**) with different fillers particle size.

**Figure 3 polymers-13-02284-f003:**
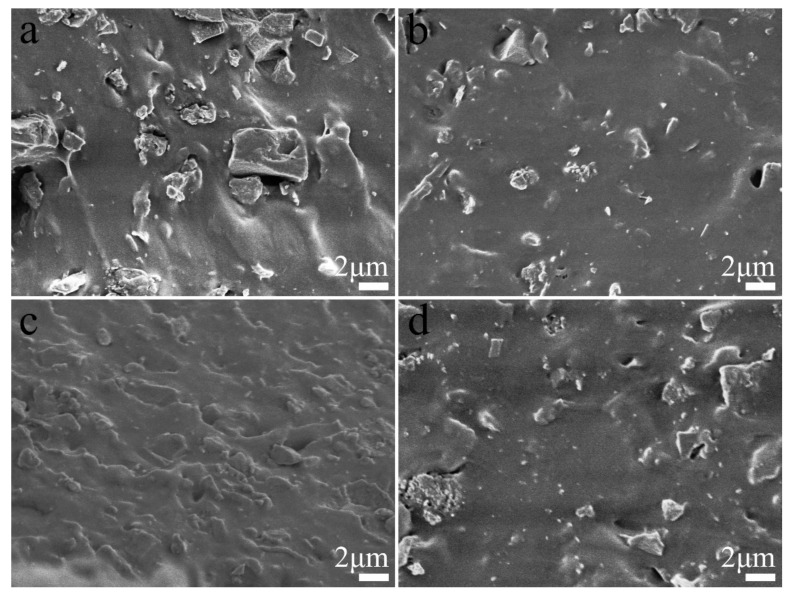
SEM images of the brittle fracture surfaces (**a**) SiCB-NR/BR, (**b**) M_1_SiCB-NR/BR, (**c**) SiCB-MNR_3_/BR, and (**d**) M_2_SiCB-NR/BR.

**Figure 4 polymers-13-02284-f004:**
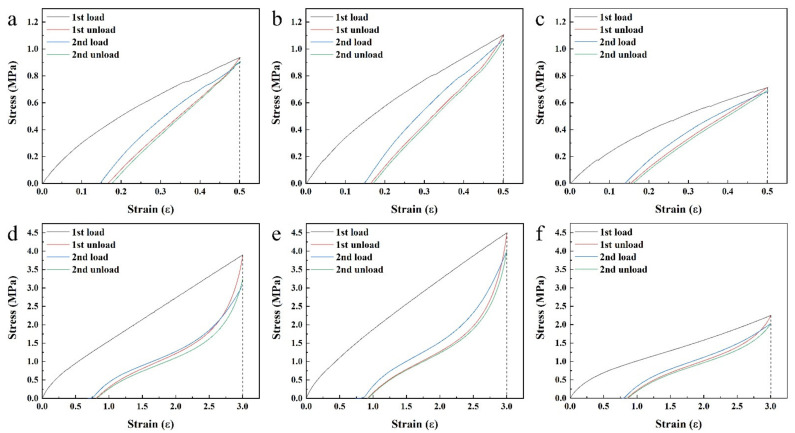
Stress–strain response of SiCB-NR/BR (**a**,**d**), SiCB-MNR_3_/BR (**b**,**e**) and M_1_SiCB-NR/BR (**c**,**f**) subjected to uniaxial cyclic tension with different strain (**a**–**c**, *ε* = 0.5; **d**–**f**, *ε* = 3.0).

**Figure 5 polymers-13-02284-f005:**
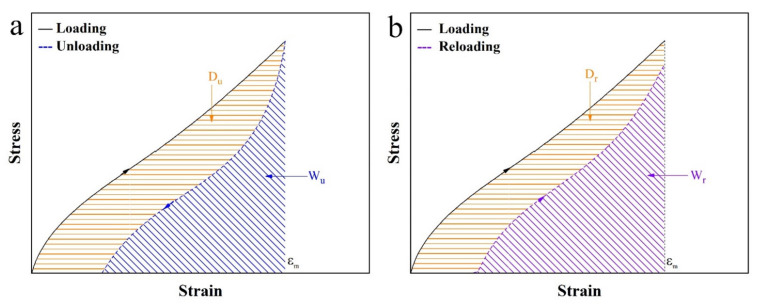
Schematics for the evaluations of energy dissipation based on (**a**) loading–unloading and (**b**) loading–reloading curves with the same maximum strain of *ε_m_*. *W* and *D* are the stored and dissipated energies, respectively. The subscripts u and r denote unloading and reloading, respectively.

**Figure 6 polymers-13-02284-f006:**
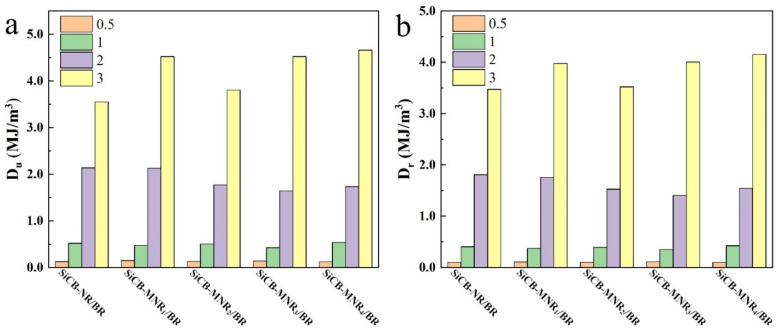
Energy dissipation of SiCB-MNR_x_/BR composites with different crosslinking densities at different *ε* during the (**a**) unloading and (**b**) reloading processes.

**Figure 7 polymers-13-02284-f007:**
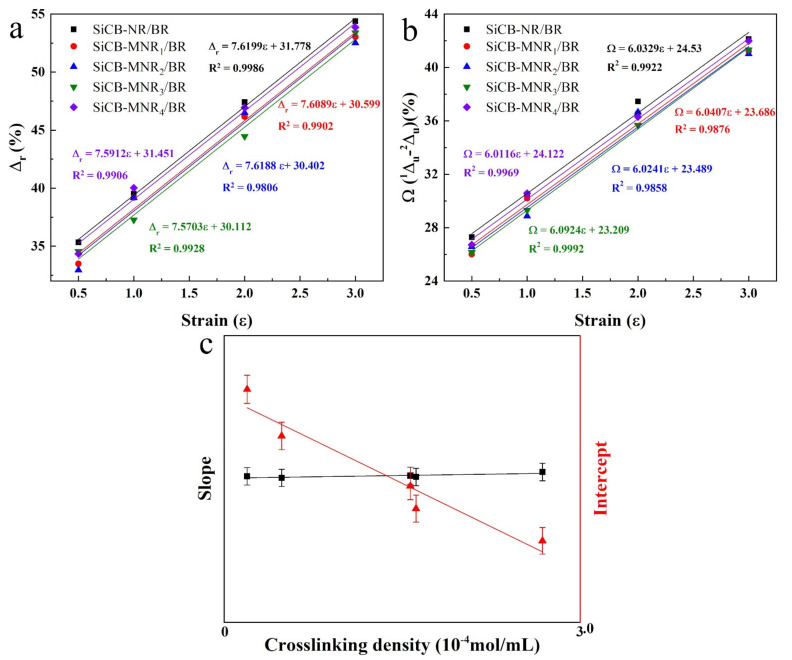
(**a**) The dissipation factor (*Δr*), (**b**) the difference of dissipation factor (*Ω*) as a function of *ε*m for SiCB-MNRx/BR composites with different crosslinking densities, and (**c**) schematic for the effect of crosslinking degree of polymers on Mullins effect.

**Figure 8 polymers-13-02284-f008:**
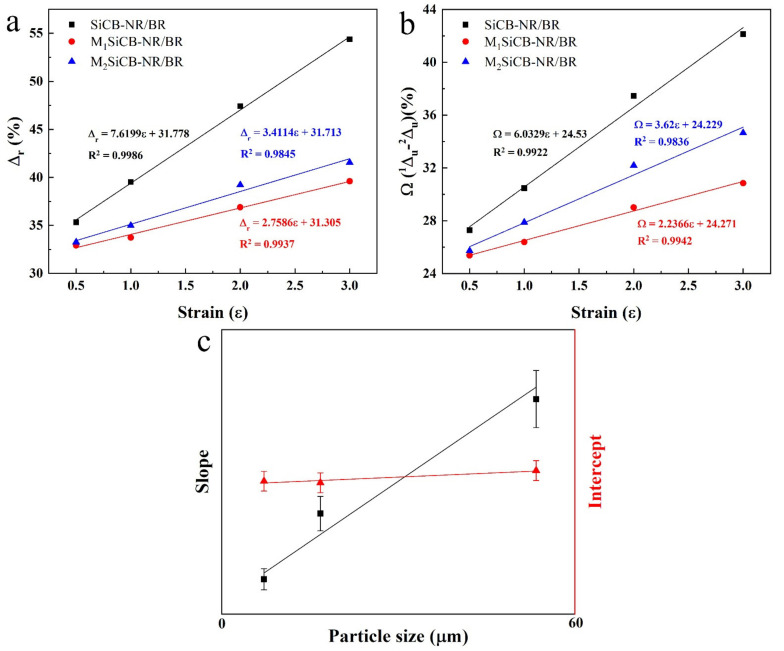
(**a**) The dissipation factor (*Δ_r_*), (**b**) the difference of dissipation factor (*Ω*) as a function of *ε_m_* for SiCB-NR/BR composites with different fillers particle size, and (**c**) schematic for the effect of filler size on Mullins effect.

**Figure 9 polymers-13-02284-f009:**
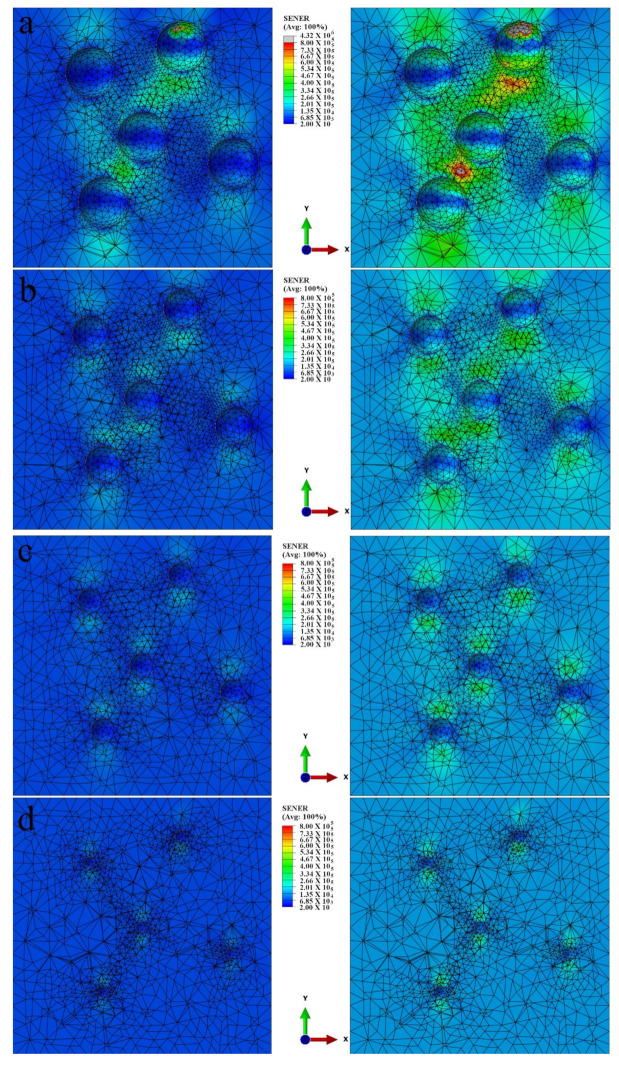
Contour plots of strain energy density distributions in the uniaxial specimen for model NR/BR composites filled with different particle size with diameter of (**a**) r = 0.4 mm, (**b**) r = 0.3 mm, (**c**) r = 0.2 mm, (**d**) r = 0.1 mm, at different strain level from 25% to 35%.

**Table 1 polymers-13-02284-t001:** Recipes for preparation of SiCB-NR/BR.

Composites	Filler	NR (phr)	BR (phr)	MA (phr)
SiCB-NR/BR	SiCB	45	55	0
M_1_SiCB-NR/BR	M_1_SiCB	45	55	0
M_2_SiCB-NR/BR	M_2_SiCB	45	55	0
SiCB-MNR_1_/BR	SiCB	45	55	1
SiCB-MNR_2_/BR	SiCB	45	55	2
SiCB-MNR_3_/BR	SiCB	45	55	3
SiCB-MNR_4_/BR	SiCB	45	55	4

**Table 2 polymers-13-02284-t002:** Crosslinking densities determined from both equilibrium swelling and tensile testing.

Composites	[*ν*]*_es_* (10^−4^ mol/mL)	[*ν*]*_ts_* (10^−4^ mol/mL)
SiCB-NR/BR	2.460	2.751
SiCB-MNR_1_/BR	2.602	2.909
SiCB-MNR_2_/BR	2.607	3.004
SiCB-MNR_3_/BR	2.717	3.148
SiCB-MNR_4_/BR	2.490	2.680

## Data Availability

The data presented in the study are available on request from the corresponding author.

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
