# Peer review of "The Influence of Filler Size and Crosslinking Degree of Polymers on Mullins Effect in Filled NR/BR Composites"

_polymers, 2021, doi:10.3390/polym13142284_

Round 1

Reviewer 1 Report

  1. The current study investigates the effect of filler size and the degree of crosslinking in polymers on the mullins effect in rubber-based composites. For this, the authors consider two factors which influence elastomers and analyse them and study the dissipation ratios as a function of developed strain under cyclic loading. The authors find that the ratios increased with filler size. The authors also appear to do some modelling but not clear what is it exactly about.
  2. The abstract although is written to good level, it is kind of confusing, first you mention modelling briefly but it is not clear what is the purposed of the modelling.
  3. Please consider reviewing the abstract and highlight the novelty, major findings and conclusions.
  4. The title of the manuscript does not reflect that there was any modelling done in the paper, please consider revising the title to account for that.
  5. The literature review is short and can be expanded, discuss what previous studies similar or close to this one did in the past, mention what were their findings and explain how your study brings new knowledge and difference to the field.
  6. Also, after line 72, the authors should attempt to answer the following question: What is the research gap did you find from the previous researchers in your field? Mention it properly. It will improve the strength of the article.
  7. In the materials and methods section, the authors should add the following:
  8. The mechanical and thermal properties (if available) for the rubber materials used in the study.
  9. Images and figures or the fabricated material, test equipment used in the study and any other useful figures which can give more insight about the experimental work done here.
  10. The authors are encouraged to add a list of nomenclature for all the symbols and Greek letters reported in this work at the end of the manuscript.
  11. Where is the discussion about the model? Is it mathematical or numerical, I am confused now. Looking at the end of the manuscript there is some FE done but there is no explanation about it before the results and discussion section?
  12. The authors must add full details of the FE model, boundary conditions, mesh convergence study, element type, number…etc.
  13. Also, what is the overall purpose of the numerical study, what is the aim, the added value? Was it just performed because everybody is using finite elements? This was not explained (typically a model is built, then validated with data from a test campaign and afterwards used for additional studies to saving experimental efforts).
  14. There is no explanation at all concerning the verification and validation of the model, it was just built and used for strain energy density distributions simulations.
  15. Line 241 why the authors did not mention they measured these metrics “stored elastic energy in loading and reloading processes (W0 and Wr),” in the abstract?
  16. Line 262 and 264 check referencing style there (superscript is used).
  17. Figure 5 why the stress strain graphs are missing data (numbers) in x and y axis?
  18. 338-339 how about previous studies, did they also report similar results as yours or different (particle size and cross linking degree), in either case, the authors should expand upon this point and compare their work with previous literature.
  19. Line 351-352 “Composites with larger particle size show a higher strain energy density of…” why? What is causing this phenomenon, please discuss and support with references if necessary. Also is this always the case with increase in particle size or does it vary if larger particle sizes were used?
  20. I don’t see the point of the FE model graphs, please either remove them or provide proper discussion on them and correlate them with the experimental work you did.

Reviewer 2 Report

The present manuscript entitled “The influence of filler size and crosslinking degree of polymers on Mullins effect in filled NR/BR composites” authored by Miaomiao Qian et al. describes the influence of two key factors, crosslinking degree of matrix and size of filler, on the Mullins effect were systematically investigated. Furthermore, the current study designates the Δr and W associated with energy dissipation from different composites subjected to uniaxial cyclic tensile testing are used to evaluate the effect of filler particle size and polymers crosslinking degree on the Mullins effect.  The text does not contain major language mistakes. The objective and justification of the work are clear, and the experimental work is significant. The study is very accurate and adequate, and thus, I recommend it for publication. However, certain Minor issues are detailed below which need to be addressed before its final acceptance in the Polymers.

Comment 1:  There are some typographical errors in the manuscript text, so authors need to correct them in the revised manuscript.

Comment 2: The introduction is well written, appropriate information is provided. However, include some more recent year’s literature in the introduction section to strengthen their work 

Comment 3: Figures 1a quality is very poor, so improve the Figure 1a resolution.

Comment 4: The SEM images of the brittle fracture surfaces explanation should be discussed wider and compared with the other studies.

Comment 5:  Observed some copied texts and sentences in your manuscript at the step of full-text plagiarism check. In order to avoid confusion and misunderstanding of the readership. I suggest you rephrase those sentences Lines 28-36, 86-99, 104-106, 109-112, 126-131, and 143-159 (the plagiarism report is attached for your reference as well).

Round 2

Reviewer 1 Report

All questions answered paper can be accepted